# Biometric characteristics of winter rape plants (Brassica napus L.) before harvest in the soil and climatic conditions of north-eastern Poland

Anna Sikorska[1]*, Marek Gugała[2], Krystyna Zarzecka[2], Iwona Mystkowska[3], Agnieszka Ginter[2], Pavol Findura[4], Miroslav Pristavka[5]

1 Department of Agriculture, Ignacy Mościcki University of Applied Sciences in Ciechanów, Ciechanów, Poland, 2 Faculty of Agrobioengineering and Animal Husbandry, University of Natural Sciences and Humanities in Siedlce, Siedlce, Poland, 3 Department of Dieteties, Pope John Paul II State School of Higher Education in Biała Podlaska, Biala Podlaska, Poland, 4 Department of Machines and Production Biosystems, Faculty of Engineering, Slovak University of Agriculture in Nitra, Nitra, Slovakia, 5 Department of Quality and Engineering Technology, Faculty of Engineering, Slovak University of Agriculture in Nitra, Nitra, Slovakia

* anna.sikorska@puzim.edu.pl

**Data Availability Statement:** All relevant data are within the paper and its Supporting Information.

## Abstract

The research was based on a field experiment carried out in the Agricultural Experimental Station in the climatic and soil conditions of north-eastern Poland. The factors of the experiment were: I–morphotypes of winter oilseed rape: population, restored hybrid with a traditional type of growth, restored hybrid with a semi-dwarf type of growth. II–methods of using preparations: variant (1)—control–without using preparations; variant (2)–an organic preparation containing microorganisms as well as micro and macro elements (Ugmax); variant (3)–a biostimulant containing 13.0% of $P_2O_5$ and 5.0% of potassium oxide ($K_2O$); variant (4)–a biostimulant containing silicon. The objectives of study was to determine the effect of preparations containing microorganisms as this well as micro and macro-elements, phosphorus and potassium and silicon on the morphometric features of plants, such as: the height of the first fruit-bearing lateral branching on the main shoot, the thickness of the stem at the base, number of productive branches and siliques on the plant, the length of the pods, plant height before harvesting. The organic preparation containing microorganisms as well as micro and macro-elements, applied in the autumn before sowing seeds and in the spring after the start of vegetation, had the most beneficial effect on the biometric characteristics of plants before harvesting.

## Introduction

The use of simplifications in tillage and crop rotation results in the impoverishment of the soil environment in beneficial microorganisms. Authors who have presented attractive results in this field [1, 2] believe that continuous and excessive use of chemical fertilizers and plant protection products causes ecological and health hazards, and also worsens the physical and chemical condition of the soil, contributing as a result to reduction in crop yield and quality.

**Funding:** The results of the research carried out under the research theme No. 32/20/B were financed from the science grant granted by the Ministry of Science and Higher Education. The funders had no role in study design, data collection and analysis, decision to publish, or preparation of the manuscript.

**Competing interests:** The authors have declared that no competing interests exist.

Biostimulants and bioprotectants are natural preparations that have gained a lot of interest in recent years due to their role in improving plant growth and yield by reducing the impact of abiotic and biotic stresses [3]. Du Jardin [4] gives a division of biostimulants into individual categories: humic acid and fulvic acid, proteolytic substances, seaweed extracts, chitosan, inorganic compounds, as well as beneficial fungi and bacteria.

According to many researchers, such as: Iwaishi [5]; Stielow [6]; Yamada and Xu [7]; Boligłowa [8], Rutkowska and Pikuła [9], application of the biopreparation/biofertilizer has a positive effect on increasing the biological activity of the soil, humus content, reducing putrefactive processes, detoxifying soil contaminated with pesticides, improving the assimilability of compounds difficult to access for plants, increasing the photosynthesis effect, inhibiting the development of phytopathogens and improving the quality of crop yields. Vessey [10] emphasizes that numerous species of soil bacteria that develop in the rhizosphere of plants, but which can grow in, on or around plant tissues, stimulate plant growth by multiple mechanisms.

Therefore, more and more often in agricultural practice, e.g. biofertilizer, which accelerate the decomposition of mineral and organic fertilizers and increase the availability of nutrients for plants [11, 12].

The research hypothesis was adopted that natural preparations of organic and inorganic origin may have a beneficial effect on the biometric features of winter oilseed rape plants. The objectives of study was to determine the effect of preparations containing microorganisms as this well as micro and macro-elements (variant 2), phosphorus and potassium (3) and silicon (4) on the morphometric features of plants, such as: the height of the first fruit-bearing lateral branching on the main shoot, the thickness of the stem at the base, number of productive branches and siliques on the plant, the length of the pods, plant height before harvesting. The paper presents a varied reaction of winter oilseed rape morphotypes to the applied preparations and their effect in individual years of research.

## Material and methods

### Experiment location

The research was based on a field experiment carried out in 2018–2021 at the Agricultural Experimental Station, in the climatic and soil conditions of north-eastern Poland. The experiment was set up in a split-plot system in three repetitions. Conducting the field experiment did not require obtaining any permits for field work, because the Zawady Agricultural Experimental Station is part of the University of Natural Sciences and Humanities in Siedlce. Field experiments have been conducted in the Zawady experimental station for over 40 years.

### Experiment factors

The main factors of the experiment were: three morphotypes of winter rape and types of preparations used. The experiment also examined the impact of climatic conditions prevailing in the years of the experiment on the biometric characteristics of plants before harvesting.

### I–morphotypes of winter rape

- population morphotype (Chrobry variety),

- hybrid morphotype with a traditional type of growth (PT 271)

- hybrid morphotype with a semi-dwarf type of growth (PX 113)

Table 1. Preparations used in the experiment.

| METHODS OF USING PREPARATIONS | Time of application of preparations in the BBCH phase | DOSE |
|---|---|---|
| variant 1 –control object | without using the preparations | - |
| variant 2- organic preparation (Ugmax) | I–in autumn before sowing seeds | 0.9 $dm^3 \cdot ha^{-1}$ |
| | II–in spring after the start of vegetation: the beginning of the development of side shoots (BBCH 21–36) | |
| variant 3 –biostimulant containing 13.0% of $P_2O_5$ and 5.0% of $K_2O$ | I–in autumn, 4–6 leaves phase (BBCH 13–15) | 1.0 $dm^3 \cdot ha^{-1}$ |
| | II–in spring after the start of vegetation (BBCH 28–30) | |
| variant 4 –biostimulant containing silicon | I–in autumn, 4–6 leaves phase (BBCH 13–15) | 0.5 $dm^3 \cdot ha^{-1}$ |
| | II–dense green flower bud in spring (BBCH 51) | |

## II–methods of using the preparations

Table 1 describes the preparations used in the experiment, the dates of their application and the dose. One of the biofertilizers used in the experiment was the Ugmax preparation. It is a microbiological preparation consisting of yeast, lactic acid bacteria, photosynthetic bacteria, *Azotobacter*, *Pseudomonas* and *Actinomycetes* and nutrients such as: potassium (3500 mg·$dm^{-3}$), nitrogen (1200 mg·$dm^{-3}$), sulphur (1000 mg·$dm^{-3}$), phosphorus (500 mg·$dm^{-3}$), sodium (200 mg·$dm^{-3}$), magnesium (100 mg·$dm^{-3}$), zinc (20 mg·$dm^{-3}$) and manganese (0,3 mg· $dm^{-3}$).

Another preparation used in the experiment was a root system development stimulator, accelerating its regeneration, improving the uptake of minerals from the soil, containing 13.0% of $P_2O_5$ and 5.0% of $K_2O$.

In the last experimental facility, an anti-stress agent containing an active and bio-available form of silicon (Optisil) was used to increase plant tolerance to unfavourable cultivation conditions and reduce biotic stresses caused by pathogens and pests.

## Soil conditions

The experiment was carried out on soil classified to the Haplic Luvisol group, sandy, very good rye soil complex, valuation class IVa [13]. The pH of the soil was slightly acidic and ranged from 5.68 to 5.75 in the years of research. The soil was characterized by low abundance in assimilable forms of phosphorus and medium assimilability in potassium and magnesium.

## Fertilization

The winter triticale was the forecrop for winter rape in individual years of the study. Table 2 shows the applied mineral fertilization.

## Sowing

Winter rape was sown at a row spacing of 22.5 cm, maintaining the density of 45 plants per $m^{-2}$. The sowing was carried out at the optimal time recommended for this region (from 10 to 15 August).

## Chemicals

Chemicals against weeds, diseases and pests was applied in accordance with the recommendations of good agricultural practice. Treatments with the use of plant protection products were carried out in accordance with the recommendations for their use, so as to ensure the assumed effectiveness at the minimum necessary dose, taking into account local conditions and the possibility of combating with mechanical methods (Table 3).

**Table 2. Mineral fertilization used in the experiment.**

| Date of fertilization | Type of fertilization | Doses |
|---|---|---|
| before sowing | phosphorus-potassium fertilization | 40 kg P·ha$^{-1}$ |
| | | 110 kg K·ha$^{-1}$ |
| | | first dose 40 kg N·ha$^{-1}$ |
| during autumn growth and development | Lubofos | dose 600 kg containing: |
| | | 21 kg N·ha$^{-1}$ |
| | | 26.4 kg P·ha$^{-1}$ |
| | | 92.1 kg K·ha$^{-1}$ |
| | | 34.8 kg S·ha$^{-1}$ |
| | | 1.2 kg B·ha$^{-1}$ |
| | ammonium sulphate | 55.9 kg·ha$^{-1}$ (19 kg N·ha$^{-1}$), |
| | triple superphosphate | 29.6 kg·ha$^{-1}$ (13.6 kg P·ha$^{-1}$) |
| | potassium salt | 29 kg·ha$^{-1}$ (17.9 kg K·ha$^{-1}$) |
| in spring before vegetation starts (BBCH 28–30) | ammonium nitrate | 255.5 kg·ha$^{-1}$ (86.9 kg N·ha$^{-1}$) |
| | ammonium sulphate | 62.5 kg·ha$^{-1}$ (13.1 kg N·ha$^{-1}$ +15 kg S·ha$^{-1}$) |
| at the beginning of budding (BBCH 50) | ammonium nitrate | 176.5 kg·ha$^{-1}$ (60 kg N·ha$^{-1}$) |

## Harvest

Rapeseed was harvested at the stage of full maturity (BBCH 79), in two stages in the first and second decade of July.

## Evaluation of yielding components

Immediately before harvesting (BBCH 86–87), the following elements of the yield structure were determined on a sample of 20 plants from each plot:

- the height of the first fruit-bearing lateral branching on the main shoot (cm),

- thickness of the stem at the base (mm),

- number of productive branches (pcs.),

- number of siliques on the plant (pcs.),

**Table 3. Type of chemical protection treatments.**

| Active substance in the agent used | Dose of the agent used | Development phases | acc. to the BBCH scale |
|---|---|---|---|
| **HERBICIDES** | | | |
| clomazone | 0.25 dm$^3$·ha$^{-1}$ | immediately after sowing on the soil | 00 BBCH |
| fluazifop-P-butyl | 2.0 dm$^3$·ha$^{-1}$ | phase of 3–4 leaves | 13–14 BBCH |
| **INSECTICIDES** | | | |
| thiacloprid deltamethrin | 0.6 dm$^3$·ha$^{-1}$ | 1 treatment–growth of the main shoot, | 30 BBCH |
| | | 2 treatment–development of flower buds | 50–58 BBCH |
| | | 3 treatment—flowering | 60–69 BBCH |
| **FUNGICIDES** | | | |
| tebuconazole | 0.75 dm$^3$·ha$^{-1}$ | phase of 4–8 leaves of rape | 14–18 BBCH |
| fluopyram prothioconazole | 1.0 dm$^3$·ha$^{-1}$ | beginning of flowering | 61 BBCH |
| prochloraz | 1.0 dm$^3$·ha$^{-1}$ | fall phase of the first flower petals | BBCH |

■ the length of the pods (cm),

■ plant height before harvesting(cm).

## Statistical analysis

The results of the research were statistically processed using the analysis of variance. The significance of the sources of variability was tested with the Fischer-Snedecor "F" test, and the assessment of the significance of differences at the significance level of α = 0.05 between the compared means was tested with multiple Tukey intervals. Statistical calculations were made based on a proprietary algorithm written in Excel in accordance with a mathematical model.

## Climatic conditions

In the years of the research, differences in terms of climate were demonstrated (Tables 4 and 5). In the second year of the study (2018–2021), the highest rainfall was recorded, which was 419.0 mm on average, and the highest average air temperature was 10.1˚C. It was a very wet growing season (K = 2.68), with quite dry, dry, very dry months, and the period from May to July was extremely wet. In the 2018–2019 growing season, the lowest rainfall was recorded (average 244.0 mm), and the average air temperature in this growing season was 9.7˚C. The last year of the study was characterized by the lowest average air temperature of 9.3˚C on

**Table 4. The value of the Sielianinow hydrothermal coefficient.**

| | Value of the Sielianinow hydrothermal coefficient * | | | | | | | | |
| --- | --- | --- | --- | --- | --- | --- | --- | --- | --- |
| | VIII | IX | X | III | IV | V | VI | VII | Mean |
| I year of research (2018–2019) | 1.19 | 1.72 | 2.42 | 3.13 | 0.60 | 4.49 | 1.68 | 1.60 | 2.10 |
| II year of research (2019–2020) | 2.20 | 1.22 | 0.89 | 1.31 | 0.70 | 5.42 | 6.14 | 3.56 | 2.68 |
| III year of research (2020–2021) | 0.90 | 2.50 | 4.39 | 3.55 | 6.36 | 2.38 | 1.66 | 2.20 | 2.99 |

* Coefficient value [14]: Extremely dry (ss) k≤0.4; Very dry (bs) 0.4–0.7; Dry (s) 0.7–1.0; Rather dry (ds) 1.0<k≤1.3; Optimal (o) 1.3<k≤1.6; Rather wet (dw) 1.6<k≤2.0; Wet (w) 2.0<k≤2.5; Very wet (bw) 2.5<k≤3.0; Extremely wet (sw) k>3.0

**Table 5. Climatic data 2019–2021 (Agricultural Experimental Station, Poland).**

| | Precipitation (mm) | | | | Air temperature (˚C) | | | |
| --- | --- | --- | --- | --- | --- | --- | --- | --- |
| | I year | II year | III year | Multiyear total (1996–2010) | I year | II year | III year | Multiyear mean (1996–2010) |
| VIII | 24.5 | 43.9 | 18.2 | 59.9 | 20.6 | 19.9 | 20.2 | 18.5 |
| IX | 27.4 | 17.4 | 38.8 | 42.3 | 15.9 | 14.2 | 15.5 | 13.5 |
| X | 23.3 | 9.5 | 52.7 | 24.2 | 9.6 | 10.7 | 12.0 | 7.9 |
| XI | 9.8 | 17.8 | 34.0 | 20.2 | 3.3 | 6.1 | 5.0 | 4.0 |
| XII | 9.0 | 29.1 | 34.0 | 18.6 | 0.4 | 2.9 | -1.0 | -0.1 |
| I | 7.9 | 12.9 | 22.6 | 19.0 | -3.0 | 1.9 | -1.9 | -3.2 |
| II | 4.7 | 26.8 | 10.4 | 16.0 | 2.2 | 2.9 | -2.5 | -2.3 |
| III | 15.0 | 5.9 | 9.6 | 18.3 | 4.8 | 4.5 | 2.7 | 2.4 |
| IV | 5.9 | 6.0 | 42.0 | 33.6 | 9.8 | 8.6 | 6.6 | 8.0 |
| V | 59.8 | 63.5 | 29.5 | 58.3 | 13.3 | 11.7 | 12.4 | 13.5 |
| VI | 35.9 | 118.5 | 33.8 | 59.6 | 21.4 | 19.3 | 20.4 | 17.0 |
| VII | 29.7 | 67.7 | 50.0 | 57.5 | 18.5 | 19.0 | 22.7 | 19.7 |
| Mean | 244.0 | 419.0 | 375.6 | 427.5 | 9.7 | 10.1 | 9.3 | 8.2 |

average. In terms of the calculated Sielianinov coefficient, it was a very wet year of research (K = 2.99).

## Results and discussion

Based on the research, it was shown that the population morphotype was distinguished by the **highest plant height**. This value was higher by 13.7 cm on average compared to the long-stemmed hybrid (Tables 6 and 7). However, other authors [15, 16] obtained the same height in a restored hybrid with a traditional growth type and population variety. The types of preparation used significantly increased the value of this feature compared to the control variant (Tables 6 and 7). Plants of all tested cultivars were the highest after application of the organic preparation containing microorganisms as well as micro and macro-elements, significantly lower after the application of the preparation with silicon, and the lowest after supplementation with the preparation with 13.0% of $P_2O_5$ and 5.0% of $K_2O$. It should be emphasized that among the tested cultivars, the largest significant increase of 8.4 cm on average was shown in the population morphotype after the application of the Ugmax preparation. In the half-dwarf hybrid, the same value of this feature was found in the objects where the preparation with phosphorus and potassium and silicon were applied. Similar conclusions were reached by Matysiak and Miziniak [17], who showed that winter rape plants after application of the mixture of Kelpak and Lithovit in the BBCH 32 phase, and twice Lithovit (BBCH 32 and 67) were characterized by an average height of 10–11% higher compared to control plants. Similar results were obtained by Sikorska et al. [15] and Przybysz et al. [18], after using the Asahi SL biostimulant. Sikorska et al. [16] after applying amino acids, sulphur and boron, noted a significant increase in plant height before harvest from 2.3 to 5.5 cm, but the differences between the variants were statistically insignificant. Nowak and Wenda-Piesik [19] also found a beneficial effect of a preparation containing 19 amino acids on plant growth. Dautartė et al. [20] after using a liquid bioorganic Raskila preparation containing a complex of macro- and trace elements, humic substances, fulvic acids, nitrogen, potassium, phosphorus, phyto-vitamins and beneficial soil microorganisms, they obtained a significant increase in plant height, on average, from 118.16 to 127.64 cm.

Own research showed that the **height of the first productive branching** was the highest in the population morphotype. The highest value of this feature was also shown on object 2, where an organic preparation was used (Tables 6 and 7). This value was higher on average by 3.3 cm compared to the control variant. Differences between the object where phosphorus and potassium and silicon were used were statistically insignificant in all tested morphotypes. The semi-dwarf hybrid had lower productive branching compared to the long-stem cultivar, while after applying the organic preparation, both hybrids showed the same reaction to the applied limited preparation.

The research showed that the hybrid with the traditional type of growth had the **largest number of productive branches,** while the population form and the semi-dwarf hybrid were characterized by the same value of this feature. Significantly the highest value of this feature was recorded after the application of the organic preparation, and the lowest after the application of preparations containing phosphorus and potassium. It should be noted that the tested morphotypes had the same number of productive branches after application of Ugmax (Tables 6 and 7). Similarly, Passandideh et al. [21] after application of amino acids and Dautartė et al. [20] after application of organic preparations obtained a significant increase in the value of this feature.

**The number of siliques per plant** was the highest in the long-stemmed hybrid variety, while the population form and the semi-dwarf hybrid had the same value of this feature. After

**Table 6. Biometric characteristics of plants depending on the factors of experience.**

| Methods of using preparations | | Cultivars | | | Mean |
|---|---|---|---|---|---|
| | | population | restored hybrid with a traditional type of growth | restored hybrid with a semi-dwarf type of growth | |
| **Plant height (cm)** | | | | | |
| 1. | Variant control | 133.7 | 120.7 | 119.4 | **124.6** |
| 2. | Organic preparation containing microorganisms as well as micro and macro elements | 142.1 | 127.7 | 123.4 | **131.1** |
| 3. | Biostimulant containing 13.0% of $P_2O_5$ and 5.0% of potassium oxide ($K_2O$) | 137.0 | 123.4 | 121.0 | **127.4** |
| 4. | Biostimulant containing silicon | 139.2 | 131.1 | 122.0 | **128.6** |
| | **Mean** | **138.0** | **124.3** | **121.5** | **-** |
| | | | | $LSD_{0.05}$ for: | |
| | | | | *cultivars* | 0.6 |
| | | | | *methods of using preparations* | 0.8 |
| | | | | *interaction*: *cultivars x methods of using preparations* | 1.4 |
| **Height of the first productive branching (cm)** | | | | | |
| 1. | Variant control | 45.4 | 38.0 | 36.8 | **40.1** |
| 2. | Organic preparation containing microorganisms as well as micro and macro elements | 49.1 | 40.9 | 40.2 | **43.4** |
| 3. | Biostimulant containing 13.0% of $P_2O_5$ and 5.0% of potassium oxide ($K_2O$) | 47.9 | 40.1 | 38.9 | **42.3** |
| 4. | Biostimulant containing silicon | 49.0 | 40.3 | 39.1 | **42.8** |
| | **Mean** | **47.8** | **39.8** | **38.8** | **-** |
| | | | | $LSD_{0.05}$ for: | |
| | | | | *cultivars* | 0.5 |
| | | | | *methods of using preparations* | 0.7 |
| | | | | *interaction*: *cultivars x methods of using preparations* | 1.2 |
| **Number of productive branches (pcs.)** | | | | | |
| 1. | Variant control | 3.9 | 4.2 | 4.0 | **4.0** |
| 2. | Organic preparation containing microorganisms as well as micro and macro elements | 5.1 | 5.1 | 5.0 | **5.1** |
| 3. | Biostimulant containing 13.0% of $P_2O_5$ and 5.0% of potassium oxide ($K_2O$) | 4.1 | 4.5 | 4.1 | **4.2** |
| 4. | Biostimulant containing silicon | 4.5 | 4.7 | 4.7 | **4.6** |
| | **Mean** | **4.4** | **4.6** | **4.4** | **-** |
| | | | | $LSD_{0.05}$ for: | |
| | | | | *cultivars* | 0.1 |
| | | | | *methods of using preparations* | 0.1 |
| | | | | *interaction*: *cultivars x methods of using preparations* | 0.2 |
| **Number of siliques per plant (pcs.)** | | | | | |
| 1. | Variant control | 130.6 | 136.8 | 128.9 | **132.1** |
| 2. | Organic preparation containing microorganisms as well as micro and macro elements | 143.6 | 150.2 | 144.2 | **146.0** |
| 3. | Biostimulant containing 13.0% of $P_2O_5$ and 5.0% of potassium oxide ($K_2O$) | 131.2 | 133.1 | 132.0 | **132.1** |
| 4. | Biostimulant containing silicon | 139.3 | 145.8 | 139.6 | **141.6** |
| | **Mean** | **136.2** | **141.5** | **136.2** | **-** |
| | | | | $LSD_{0.05}$ for: | |
| | | | | *cultivars* | 1.7 |
| | | | | *methods of using preparations* | 1.5 |

*(Continued)*

**Table 6.** (Continued)

| Methods of using preparations | | population | restored hybrid with a traditional type of growth | restored hybrid with a semi-dwarf type of growth | Mean |
|---|---|---|---|---|---|
| | | | **Cultivars** | | |
| | | | *interaction: cultivars x methods of using preparations* | | 2.6 |
| **Length of the pods (cm)** | | | | | |
| 1. | Variant control | 7.0 | 7.2 | 6.9 | **7.0** |
| 2. | Organic preparation containing microorganisms as well as micro and macro elements | 7.8 | 8.1 | 7.8 | **7.9** |
| 3. | Biostimulant containing 13.0% of $P_2O_5$ and 5.0% of potassium oxide ($K_2O$) | 7.0 | 7.5 | 7.0 | **7.2** |
| 4. | Biostimulant containing silicon | 7.5 | 7.7 | 7.5 | **7.6** |
| **Mean** | | **7.3** | **7.6** | **7.3** | - |
| | | | | **$LSD_{0.05}$ for:** | |
| | | | | *Cultivars* | 0.1 |
| | | | | *methods of using preparations* | 0.1 |
| | | | | *interaction: odmiany x methods of using preparations* | 0.1 |
| **Thickness of the stem at the base (mm)** | | | | | |
| 1. | Variant control | 13.80 | 13.62 | 13.91 | **13.78** |
| 2. | Organic preparation containing microorganisms as well as micro and macro elements | 15.04 | 14.39 | 15.21 | **14.88** |
| 3. | Biostimulant containing 13.0% of $P_2O_5$ and 5.0% of potassium oxide ($K_2O$) | 14.82 | 13.98 | 14.11 | **14.30** |
| 4. | Biostimulant containing silicon | 14.68 | 13.97 | 14.24 | **14.30** |
| **Mean** | | **14.59** | **13.99** | **14.37** | - |
| | | | | **$LSD_{0.05}$ for:** | |
| | | | | *cultivars* | 0.17 |
| | | | | *methods of using preparations* | 0.24 |
| | | | | *interaction: cultivars x methods of using preparations* | 0.42 |

the application of the organic preparation, the number of siliques on the plant increased by an average of 14.0 pcs compared to the control variant, and after the application of the preparation containing phosphorus and potassium, this value was the same as on the object where no preparations were used. It should be emphasized that in the hybrid morphotype with a traditional type of growth, after the application of the organic preparation, this value increased on average by 14.4 units, while the population form and the half-dwarf hybrid were characterized by the same value of this feature on this object (Tables 6 and 7). Jarecki et al. [22], after foliar application of a preparation containing Mg, B, Cu, Fe, Mn, Mo, Zn showed that it significantly increased the number of siliques on the plant compared to the control. Chwil [23] showed that the applied treatments of foliar feeding and soil fertilization increased the number of siliques on rapeseed plants of the Kana cultivar. The author emphasizes that the greatest value of this feature was caused by the application of nitrogen, phosphorus and potassium with magnesium lime.

**The length of the pods** was the greatest in the long-stemmed hybrid, while the population variety and the semi-dwarf hybrid had the same value of this feature. On the object with the organic preparation, the plants were characterized by the greatest length of siliques, but the population form and the semi-dwarf hybrid were characterized by the same value of this feature on this object (Tables 6 and 7).

**The thickness of the stem at the base** was the greatest in the population morphotype, significantly lower in the semi-catalyst hybrid, and the smallest in the long-stemmed form

**Table 7. Biometric characteristics of plants depending on years of research and morphotypes.**

| Years | Cultivars | | | Mean |
|---|---|---|---|---|
| | population | restored hybrid with a traditional type of growth | restored hybrid with a semi-dwarf type of growth | |
| **Plant height (cm)** | | | | |
| I | 131.5 | 121.5 | 120.0 | **124.3** |
| II | 142.8 | 126.8 | 122.0 | **130.5** |
| III | 139.8 | 124.6 | 122.3 | **128.9** |
| **Mean** | **138.0** | **124.3** | **121.5** | - |
| | | | LSD$_{0.05}$ for: | |
| | | | *cultivars* | 0.6 |
| | | | *years* | 0.6 |
| | | | *interaction: cultivars x years* | 1.1 |
| **Height of the first productive branching (cm)** | | | | |
| I | 40.7 | 35.7 | 34.1 | **36.8** |
| II | 57.2 | 43.4 | 42.5 | **47.7** |
| III | 45.6 | 40.3 | 39.7 | **41.9** |
| **Mean** | **47.8** | **39.8** | **38.8** | - |
| | | | LSD$_{0.05}$ for: | |
| | | | *cultivars* | 0.5 |
| | | | *years* | 0.5 |
| | | | *interaction: cultivars x years* | 0.8 |
| **Number of productive branches (pcs.)** | | | | |
| I | 3.9 | 3.9 | 3.7 | **3.8** |
| II | 5.0 | 5.4 | 5.2 | **5.2** |
| III | 4.4 | 4.6 | 4.5 | **4.5** |
| **Mean** | **4.4** | **4.6** | **4.4** | - |
| | | | LSD$_{0.05}$ for: | |
| | | | *Cultivars* | 0.1 |
| | | | *years* | 0.1 |
| | | | *interaction: cultivars x years* | 0.2 |
| **Number of siliques per plant (pcs.)** | | | | |
| I | 125.9 | 130.9 | 124.8 | **127.2** |
| II | 150.4 | 148.8 | 146.9 | **148.7** |
| III | 132.2 | 144.6 | 136.9 | **137.9** |
| **Mean** | **136.2** | **141.5** | **136.2** | - |
| | | | LSD$_{0.05}$ for: | |
| | | | *cultivars* | 1.7 |
| | | | *years* | 1.7 |
| | | | *interaction: cultivars x years* | 2.9 |
| **Length of the pods (cm)** | | | | |
| I | 6.7 | 6.9 | 6.6 | **6.7** |
| II | 8.2 | 8.5 | 7.9 | **8.2** |
| III | 7.1 | 7.6 | 7.4 | **7.3** |
| **Mean** | **7.3** | **7.6** | **7.3** | - |
| | | | LSD$_{0.05}$ for: | |
| | | | *cultivars* | 0.1 |
| | | | *years* | 0.1 |
| | | | *interaction: cultivars x years* | 0.2 |
| **Thickness of the stem at the base (mm)** | | | | |

*(Continued)*

**Table 7.** (Continued)

| | | | | |
|---|---|---|---|---|
| I | 13.63 | 13.04 | 13.66 | **13.44** |
| II | 15.74 | 15.18 | 15.48 | **15.47** |
| III | 14.39 | 13.75 | 13.97 | **14.04** |
| Mean | **14.59** | **13.99** | **14.37** | - |
| | | | LSD$_{0.05}$ for: | |
| | | | cultivars | 0.17 |
| | | | years | 0.17 |
| | | | interaction: cultivars x years | 0.29 |

**Table 8. Biometric characteristics of plants depending on the years of research and the types of preparations used.**

| Methods of using preparations | Years | | | Mean |
|---|---|---|---|---|
| | 2018–2019 | 2019–2020 | 2020–2021 | |
| **Plant height (cm)** | | | | |
| 1. Variant control | 120.4 | 127.3 | 126.2 | **124.6** |
| 2. Organic preparation containing microorganisms as well as micro and macro elements | 128.0 | 133.6 | 131.6 | **131.1** |
| 3. Biostimulant containing 13.0% of $P_2O_5$ and 5.0% of potassium oxide ($K_2O$) | 123.6 | 129.7 | 128.9 | **127.4** |
| 4. Biostimulant containing silicon | 125.2 | 131.5 | 129.0 | **128.6** |
| Mean | **124.3** | **130.5** | **128.9** | - |
| | | | LSD$_{0.05}$ for: | |
| | | | years | 0.6 |
| | | | methods of using preparations | 0.8 |
| | | | interaction: years x methods of using preparations | 1.4 |
| **Height of the first productive branching (cm)** | | | | |
| 1. Variant control | 34.5 | 46.2 | 39.5 | **40.1** |
| 2. Organic preparation containing microorganisms as well as micro and macro elements | 38.3 | 48.6 | 43.2 | **43.4** |
| 3. Biostimulant containing 13.0% of $P_2O_5$ and 5.0% of potassium oxide ($K_2O$) | 37.1 | 47.6 | 42.2 | **42.3** |
| 4. Biostimulant containing silicon | 37.4 | 48.4 | 42.6 | **42.8** |
| Mean | **36.8** | **47.7** | **41.9** | - |
| | | | LSD$_{0.05}$ for: | |
| | | | years | 0.5 |
| | | | methods of using preparations | 0.7 |
| | | | interaction: years x methods of using preparations | r.n. |
| **Number of productive branches (pcs.)** | | | | |
| 1. Variant control | 3.3 | 4.7 | 4.1 | **4.0** |
| 2. Organic preparation containing microorganisms as well as micro and macro elements | 4.4 | 5.9 | 5.0 | **5.1** |
| 3. Biostimulant containing 13.0% of $P_2O_5$ and 5.0% of potassium oxide ($K_2O$) | 3.4 | 4.7 | 4.6 | **4.3** |
| 4. Biostimulant containing silicon | 4.2 | 5.4 | 4.3 | **4.6** |
| Mean | **3.8** | **5.2** | **4.5** | - |
| | | | LSD$_{0.05}$ for: | |
| | | | years | 0.1 |
| | | | methods of using preparations | 0.1 |
| | | | interaction: years x methods of using preparations | 0.2 |
| **Number of siliques per plant (pcs.)** | | | | |
| 1. Variant control | 122.2 | 144.0 | 130.1 | **132.1** |
| 2. Organic preparation containing microorganisms as well as micro and macro elements | 134.1 | 154.9 | 148.9 | **146.0** |
| 3. Biostimulant containing 13.0% of $P_2O_5$ and 5.0% of potassium oxide ($K_2O$) | 121.9 | 143.6 | 130.8 | **132.1** |

(Continued)

**Table 8.** (Continued)

| Methods of using preparations | | Years | | | Mean |
|---|---|---|---|---|---|
| | | **2018–2019** | **2019–2020** | **2020–2021** | |
| 4. Biostimulant containing silicon | | 130.6 | 152.3 | 141.7 | **141.6** |
| | Mean | **127.2** | **148.7** | **137.9** | - |
| | | | | LSD$_{0.05}$ for: | |
| | | | | *years* | 1.7 |
| | | | | *methods of using preparations* | 1.5 |
| | | | | interaction: *years* x *methods of using preparations* | 2.6 |
| **Length of the pods (cm)** | | | | | |
| 1. Variant control | | 6.2 | 8.0 | 7.0 | **7.0** |
| 2. Organic preparation containing microorganisms as well as micro and macro elements | | 7.3 | 8.5 | 7.8 | **7.9** |
| 3. Biostimulant containing 13.0% of P$_2$0$_5$ and 5.0% of potassium oxide (K$_2$O) | | 6.3 | 7.9 | 7.2 | **7.2** |
| 4. Biostimulant containing silicon | | 7.1 | 8.4 | 7.4 | **7.6** |
| | Mean | **6.7** | **8.2** | **7.3** | - |
| | | | | LSD$_{0.05}$ for: | |
| | | | | *years* | 0.1 |
| | | | | *methods of using preparations* | 0.1 |
| | | | | interaction: *years* x *methods of using preparations* | 0.1 |
| **Thickness of the stem at the base (mm)** | | | | | |
| 1. Variant control | | 12.86 | 14.90 | 13.58 | **13.78** |
| 2. Organic preparation containing microorganisms as well as micro and macro elements | | 14.19 | 15.98 | 14.48 | **14.88** |
| 3. Biostimulant containing 13.0% of P$_2$0$_5$ and 5.0% of potassium oxide (K$_2$O); | | 13.38 | 15.42 | 14.11 | **14.30** |
| 4. Biostimulant containing silicon | | 13.34 | 15.57 | 13.98 | **14.30** |
| | Mean | **13.44** | **15.47** | **14.04** | - |
| | | | | LSD$_{0.05}$ for: | |
| | | | | *years* | 0.17 |
| | | | | *methods of using preparations* | 0.24 |
| | | | | interaction: *years* x *methods of using preparations* | 0.42 |

(Tables 6 and 7). Sikorska et al. [15] came to different conclusions, who noted the greatest thickness of the stem at the base in the restored hybrids, and significantly less in the population variety. The value of this feature was the highest after using the preparation with microorganisms. On objects 3 and 4, where phosphorus, potassium and silicon were used, respectively, the same value of this feature was obtained. Climatic conditions had a significant effect on the biometric features of plants determined before harvesting. The highest values of these features in all tested cultivars were shown in the most humid year of the research. Statistically proven interaction of years and biostimulants used proves a different effect of the biostimulant in changing climatic conditions during the research (Tables 6–8).

## Conclusions

1. The research showed that the population morphotype was characterized by the highest plant height before harvesting, the height to the first branching, the thickness of the stem at the base, and the long-stemmed hybrid had the largest number of productive branches and siliques on the plant.

2. The organic preparation containing microorganisms as well as micro and macro-elements, applied in the autumn before sowing seeds and in the spring after the start of vegetation, had the most beneficial effect on the biometric characteristics of plants before harvesting.

3. The weather conditions accompanying the vegetation of the crop has the decisive influence on the biostimulating effect of the applied preparations, in particular the amount and distribution of rainfall. The positive effect of the preparations on the biometric features of plants was revealed in the most humid year of the research.

## Supporting information

**S1 Table. Preparations used in the experiment.**
(DOCX)

**S2 Table. Mineral fertilization used in the experiment.**
(DOCX)

**S3 Table. Type of chemical protection treatments.**
(DOCX)

**S4 Table. The value of the Sielianinow hydrothermal coefficient.**
(DOCX)

**S5 Table. Climatic data 2019–2021 (Agricultural Experimental Station, Poland).**
(DOCX)

**S6 Table. Biometric characteristics of plants depending on the factors of experience.**
(DOCX)

**S7 Table. Biometric characteristics of plants depending on years of research and morphotypes.**
(DOCX)

**S8 Table. Biometric characteristics of plants depending on the years of research and the types of preparations used.**
(DOCX)

## Author Contributions

**Conceptualization:** Anna Sikorska, Marek Gugała, Krystyna Zarzecka.

**Data curation:** Marek Gugała, Krystyna Zarzecka.

**Formal analysis:** Anna Sikorska.

**Funding acquisition:** Marek Gugała.

**Investigation:** Anna Sikorska.

**Methodology:** Anna Sikorska, Marek Gugała, Krystyna Zarzecka.

**Project administration:** Iwona Mystkowska.

**Resources:** Marek Gugała, Iwona Mystkowska, Agnieszka Ginter.

**Supervision:** Marek Gugała, Pavol Findura, Miroslav Pristavka.

**Validation:** Marek Gugała.

**Writing – original draft:** Anna Sikorska.

**Writing – review & editing:** Krystyna Zarzecka.

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
