## [Decision Letter · Decision Letter 0]

15 Jun 2023

PONE-D-23-15441Biometric characteristics of winter rape plants (Brassica napus l.) before harvest in the soil and climatic conditions of north-eastern Poland, depending on the bio-stimulative preparations usedPLOS ONE

Dear Dr. SIKORSKA,

Thank you for submitting your manuscript to PLOS ONE. After careful consideration, we feel that it has merit but does not fully meet PLOS ONE’s publication criteria as it currently stands. Therefore, we invite you to submit a revised version of the manuscript that addresses the points raised during the review process.

We look forward to receiving your revised manuscript.

Kind regards,

Dharmendra Kumar Meena

Academic Editor

PLOS ONE

Journal Requirements:

https://www.mdpi.com/2077-0472/12/10/1747/htm

In your revision ensure you cite all your sources (including your own works), and quote or rephrase any duplicated text outside the methods section. Further consideration is dependent on these concerns being addressed.

"The results of the research carried out under the research theme No. 32/20/B were financed from the science grant granted by the Ministry of Science and Higher Education. "

Additional Editor Comments:

the article is recommended for major revision. Author need to improve upon the discussion part and the flow in abstract and discussion is missing. And also the cohesion between results and discussion cab be made more stronger than present one.

Reviewers' comments:

Reviewer's Responses to Questions

**Comments to the Author**

1. Is the manuscript technically sound, and do the data support the conclusions?

Reviewer #1: No

Reviewer #2: Yes

Reviewer #3: Yes

Reviewer #4: Partly

2. Has the statistical analysis been performed appropriately and rigorously? 

Reviewer #1: No

Reviewer #2: Yes

Reviewer #3: Yes

Reviewer #4: Yes

3. Have the authors made all data underlying the findings in their manuscript fully available?

Reviewer #1: Yes

Reviewer #2: Yes

Reviewer #3: Yes

Reviewer #4: Yes

4. Is the manuscript presented in an intelligible fashion and written in standard English?

Reviewer #1: No

Reviewer #2: Yes

Reviewer #3: Yes

Reviewer #4: Yes

5. Review Comments to the Author

Reviewer #1: After careful reading, I consider that the structure, logical flow, literature review and statistics used in this manuscript are not up to the standards.

I found high similarities with published literature on the internet especially the introduction section. Authors made frequent mistakes throughout the MS.

The authors would do well to refer to other peer-reviewed publications for guidelines on what is most appropriate in tables, results, and figures, and what is better placed in an appendix.

Although I am aware that there is a great effort behind the manuscript, there still are several difficult parts for publication.

Reviewer #2: It is with great pleasure that I read and evaluated the work ''Biometric characteristics of winter rape plants (Brassica napus l.) before harvest in the soil and climatic conditions of north-eastern Poland, depending on the bio-stimulative preparations used''.

This topic is long and I suggest reducing it, the work as a whole is perfect with new results, and excellent statement of the problem, the subject is relevant to the mission and purpose of the journal. The MS presents an original contribution to the literature with well-organized ideas and supporting points. In addition, there are errors in the authors citations in the text, as well as affirmative sentences which deserve to be supported by citations. I have underlined a few in the text of the manuscript. The research methodology is clear and well detailed, and the results and discussion are well presented, but some details are missing that make the MS poor if not provided by the authors. In the discussion, please expand on the main points with evidence and coherent reasoning.

Also, please follow the instructions of the journal with rigor and detailing the gray areas in order to allow readers to follow you and understand you perfectly.

Reviewer #3: I have reviewed the manuscript and it is very interesting. Dear Editor, this publication will provides new insight about the bio-stimulants and the organic preparation containing microorganisms as well as micro and macro-elements, applied in the autumn before sowing of seeds and in the spring after the start of vegetation, had the most beneficial effect on the biometric characteristics of rape plants before harvesting. I therefore, suggest the minor revision for this manuscript and recommend it for publication.

Reviewer #4: Manuscript is written well. Statistical analysis is also OK. However, suggestions are given for improvement of manuscript in all sections. You may check detailed comments in pdf of manuscript attached here with.

6. PLOS authors have the option to publish the peer review history of their article (what does this mean?). If published, this will include your full peer review and any attached files.

Reviewer #1: **Yes: **Anjana J. Atapattu

Reviewer #2: No

Reviewer #3: **Yes: **Dr. Muhammad Adnan Bukhari

Reviewer #4: **Yes: **Mudassir Aziz

---

## [Author Response · Author response to Decision Letter 0]

30 Jun 2023

The title was changed from ”Biometric characteristics of winter rape plants (Brassica napus l.) before harvest in the soil and climatic conditions of north-eastern Poland, depending on the bio-stimulative preparations used” to “Biometric characteristics of winter rape plants (Brassica napus l.) before harvest in the soil and climatic conditions of north-eastern Poland”

errors in citations have been corrected in the text and missing references to literature have been supplemented in the text. In the section on materials and methods, missing references to tables were added, years of research were specified, software used, chemical protection applied was clarified. Research results and discussion were corrected. 

The article was corrected in accordance with the editorial requirements of the journal. 

As suggested by the reviewer, the tested parameters were changed in the appropriate order, i.e. the height of the first fruit-bearing lateral branching on the main shoot, the thickness of the stem at the base, number of productive branches and siliques on the plant, the length of the pods, plant height before harvesting.

I would like to inform you, as I also included in the article, that conducting the field experiment did not require obtaining any permits for field work, because the Zawady Agricultural Experimental Station is part of the University of Natural Sciences and Humanities in Siedlce. Field experiments have been conducted in the Zawady experimental station for over 40 years. 

Major and minor experimental factors are listed as suggested,

According to the guidelines, the characteristics of the preparations used were placed in the Materials and Methods section, 

The section on chemical protection describes good agricultural practices in chemical protection of plants. 

The text indicates that the statistical calculations were made on the basis of a proprietary algorithm written in Excel in accordance with a mathematical model.

---

## [Decision Letter · Decision Letter 1]

31 Jul 2023

Charakterystyka biometryczna roślin rzepaku ozimego (Brassica napus L.) przed zbiorem w warunkach glebowo-klimatycznych północno-wschodniej Polski

PONE-D-23-15441R1

Dear Dr. ANNA

We’re pleased to inform you that your manuscript has been judged scientifically suitable for publication and will be formally accepted for publication once it meets all outstanding technical requirements.

Kind regards,

Dharmendra Kumar Meena

Academic Editor

PLOS ONE

Additional Editor Comments (optional):

I am pleased to tell you that your article can be accepted for publication in the PloSONE

Reviewers' comments:

Reviewer's Responses to Questions

**Comments to the Author**

1. If the authors have adequately addressed your comments raised in a previous round of review and you feel that this manuscript is now acceptable for publication, you may indicate that here to bypass the “Comments to the Author” section, enter your conflict of interest statement in the “Confidential to Editor” section, and submit your "Accept" recommendation.

Reviewer #2: All comments have been addressed

2. Is the manuscript technically sound, and do the data support the conclusions?

Reviewer #2: Yes

3. Has the statistical analysis been performed appropriately and rigorously? 

Reviewer #2: Yes

4. Have the authors made all data underlying the findings in their manuscript fully available?

Reviewer #2: Yes

5. Is the manuscript presented in an intelligible fashion and written in standard English?

Reviewer #2: Yes

6. Review Comments to the Author

Reviewer #2: It is with great interest that I read the manuscript ''Biometric characteristics of winter rape plants (Brassica napus L.)

before harvest in the soil and climatic conditions of north-eastern Poland''.

I notice that:

1. The authors have taken my suggestions into account,

2. The manuscript has been considerably improved, but, nevertheless there remain some small errors of punctuation in the text, I underlined some.

3. In this sentence ''In the last experimental facility, an anti-stress agent containing an active and bio-available form of silicon was used to increase plant tolerance to unfavorable cultivation conditions and reduce biotic stresses caused by pathogens and pests.'' , Please give us the name of the anti-stress agent used.

7. PLOS authors have the option to publish the peer review history of their article (what does this mean?). If published, this will include your full peer review and any attached files.

Reviewer #2: **Yes: **AIGNON Lougbegnon Hyppolite

---

## [Editor Report · Acceptance letter]

7 Aug 2023

PONE-D-23-15441R1 

Biometric characteristics of winter rape plants (Brassica napus L.) before harvest in the soil and climatic conditions of north-eastern Poland 

Dear Dr. Sikorska:

I'm pleased to inform you that your manuscript has been deemed suitable for publication in PLOS ONE. Congratulations! Your manuscript is now with our production department. 

Kind regards, 

on behalf of

Dr. Dharmendra Kumar Meena 

Academic Editor

PLOS ONE